# Supply Chain Practitioners' Perception on Sustainability: An Empirical Study

**Shaheera Haroon [1], Muhammad Wasif [2,***, Rameez Khalid [1] and Sana Khalidi [1]**

[1] Department of Management, School of Business Studies, Institute of Business of Administration, Karachi 75270, Pakistan; s.haroon.23408@khi.iba.edu.pk (S.H.); rameezkhalid@iba.edu.pk (R.K.); skhalidi@khi.iba.edu.pk (S.K.)

[2] Department of Industrial and Manufacturing Engineering, NED University of Engineering and Technology, Karachi 75270, Pakistan

* Correspondence: wasif@neduet.edu.pk

**Abstract:** The sustainability of supply chains is now one of the major global issues due to the vision of the United Nations (UN). By 2030, it is the primary focus of the UN to attain all the seventeen sustainable development goals (SDG). Hence, the primary goal of this study is to examine how practitioners think about sustainable supply chains. Five hypotheses are established to assess the perceptions of supply chain professionals. Forty-two (42) indicators are gathered from the existing literature to establish the survey instrument to collect the responses from supply chain practitioners. Finally, Structural Equation Modeling (SEM) is used to evaluate the mediation effects. It is found that social aspects are significantly impacted by the environmental aspects, as compared to the economic aspects. None of these indicators play any mediation effect, as all three are equally important for the sustainable supply chains.

**Keywords:** sustainable supply chain; green supply chain; perception; structural equation modelling; economic indicator; environmental indicator; social indicator

## 1. Introduction

For more than a decade, earth experienced record-breaking extreme weather changes. Scientists believe that if these changes are prolonged, it may have an adverse effect over the living pattern of all the species on earth. The main culprits of this environmental issue are the organizations releasing dangerous carbon dioxide gas into the atmosphere [1]. The supply chain processes of modern organizations have drastically increased the environmental pollutions, adverse changes in weather, global warming, and endangered human lives, with the release of greenhouse gases and untreated effluents, the production of non-biodegradable products, and excessive use of hazardous resources [2,3]. Due to the growing awareness about these negative impacts of industrial processes over the environment, many customers are now valuing and willing to reduce the hazardous impact over the environment. They are now showing their obligation toward environmentally friendly initiatives. These ongoing debates and concerns about the environment have changed the customers' perception, toward the products, processes, supply chains, and organizations. Manufacturers are now more inclined towards leveraging environmentally friendly procedures to win more customers. However, all these green initiatives require huge investments and return to these investments that are quite uncertain and ambiguous. Many attempts in the research domains have been made to analyze the impact of green initiatives over the organizational performance. The primary objective of such research is to evaluate whether investing in the green initiatives are enhancing the organizations' image and performance or not [1].

Sustainability is now considered as an essential element of intelligent management [4]. It transformed the organizations to perform unavoidable responsibilities and significant

attention [5]. Sustainability is an important global issue. From 113 countries, most of the managers and executives consider sustainability an essential key for planning strategies and conducting corporate activities and procedures [6]. It has changed the way of corporate thinking and now it is essential to become sustainable rather than thinking why become sustainable. This approach impacted the definition of sustainability, which is "development that meets the needs of the present without compromising the ability of future generations to meet their needs" [7]. However, this definition failed to guide how sustainability can be operationalized. The primary domain in global corporate activities are global supply chain management and sustainable supply chain management. Developing an environmentally friendly supply chain can significantly reduce the adverse effects of the organizations but may be influenced by environmental, economic, and social issues [8].

Many organizations are now adopting a concept of a green supply chain to reduce the impact of hazardous activities on the environment, to ensure social security, and to improve efficiency for the competitive advantages [9]. Nowadays, procedures are being designed to ensure the welfare of stakeholders through a reduction in greenhouse gases, reduced discharge of wastes, and low energy consumptions [10,11]. However, organizations face challenges while implementing these environmental and social friendly procedures due to ambiguous return on investment (ROI) and reduced performances [12,13]. Hence, many studies have been conducted to evaluate the impact of these green initiatives on firms' performance. Baah et al. suggested a positive relation between green supply chain initiatives and organizational performance [14]. Green et al. also recommended that green practices help in advancing environmental sustainability in manufacturing firms in the USA [15]. Zaid et al. also found the same relationship for the industries in Palestine [13]. While implementing sustainability, Carter and Rogers defined a framework based on the triple bottom line (TBL) concept [16]. Most studies presented the sustainability frameworks, based on the major elements of sustainability, which are environment, economic, and social factors [17]. Markman and Krause proposed that sustainability should be the priority of any organization, more than the economic performance, it should go over and above to achieve the sustainability [18]. There is an on-going debate that the progressive approach is insufficient, and a more holistic view is required on sustainability to achieve the ultimate sustainability goal [8].

Karmaker et al. addressed the issues of disruptions in supply chains during the COVID-19 pandemic, in context of Bangladesh. They proposed the recommendations for the sustainability in supply chain, specially the implementation of health protocols and automation [19]. He et al. presented the research to effectively sustain the supply chains through the knowledge management. They presented a comprehensive framework that integrates the knowledge management with the triple bottom line sustainability [20]. Radi et al. proposed another unique piece of research based on social media analytics for the sustainability of mobile phone supply chains. Using contents, descriptive, and sentiment analysis, they found that environmental, material, technology, and the corporate social responsibility (CSR) are the key factors for the triple bottom line sustainability [21]. Dong et al. highlighted another important issue of sustainable supply chains, related with the behavior of stakeholders. They emphasize that the other aspects of sustainability required much effort, cost, and resources, whereas behavioral change may be implemented through training, rewards, and other benefits, which may have a huge impact over sustainability [22]. Kusi-Sarpong et al. worked to highlight the importance of innovation management over the sustainability of manufacturing firms' supply chain. They concluded that innovation management is one of the key factors that influences sustainability, but it itself depends upon the financial commitments and leadership interest [23].

Various organizations view achieving environmental, social, and economic benefits by implementing green practices in some parts or in the entire supply chains. However, implementing green practices in the supply chain adds enormous costs due to poor customer awareness and government support because of inadequate regulations and environmental laws [24–27]. Despite the higher costs of the green supply chains, firms need to integrate

sustainable practices in their supply chain activities, especially logistics, due to customers' demands and increased environmental regulations for more sustainable services and products [28,29]. La Scalia et al. stated that most of the sustainable practices aim toward the economic aspect, to reduce the operational costs. They give minor attention to the environmental and social aspects of sustainability. Thus, the environmental, social, and economic aspects of sustainability need to be considered by the firms when re-engineering or designing supply chain operations [30].

The continuous emission of greenhouse gases, burning of fossil fuels, and discharging of hazardous effluents caused the climate changes, increased global warming, and resulted in more severe storms [31]. Khan et al. revealed that the poor quality of trade and transport infrastructure have a significant correlation with the emission of greenhouse gases, green energy resources, fuel consumption, political instability in the country, and health expenditure [32]. Helo et al. analyzed the potential areas of saving $CO_2$ emissions in the supply chain for the food industry, which include temperature-controlled logistics and on-line temperature control in the transportation, warehousing, and distribution aspects [33].

Regarding the environmental indicators, Hojnik et al. argued that environmental sustainability indicators include non-renewable and renewable resources, waste absorbing capacity, and resource consumption measures, especially using alternative forms of energy [34]. Stindt proposed the following two indicators for the ecological dimensions of supply chain management: resource saving and mitigating adverse environmental effects. Resource-saving comprises material intensity, recyclable products, energy efficiency, renewable energy systems, biodiversity, and land use [35]. Another study on the farming sector of Britain identified important environmental indicators for sustainable supply chains, such as improved soil and land management, improved irrigation management, improved management of natural resources, and reduced risk of environmental pollution [36]. Another study shows that sustainable packaging and environmental purchasing lead to improved compliance to environmental standards, reduced consumption of harmful/hazardous/toxic waste, and reduced energy consumption [37].

Kalenoja et al. proposed that good indicators of the energy efficiency of logistics include carbon footprint; total and specific energy consumption; consumption of water, electricity, and heat; delivery time; transport emissions; transport intensity; and load utilization rate. Values of the total energy consumption and $CO_2$ emission in each phase are to be identified [38].

Colicchia et al. investigated the strategies taken by the firms at different stages of the supply chain. Green purchasing is essential for the inbound supply chain, reducing input resources and reducing hazardous emissions and wastes at the production level. Outbound supply chains involve selecting green modes of transportation, vehicle and logistics optimization, and warehousing. It should include green sources of energy and green building practices [39]. Kim and Chai provide empirical evidence that the integration of suppliers and environmental practices are positively correlated with performance [40]. Based on the consumer perception of the sustainable practices of the supply chain in the hospitality industry, Modica et al. proposed a variety of environmental indicators, such as the purchase of greener products, greener service processes, product life extension, recycling, pollution control, and the use of environmental management systems [41]. Consumers are now paying attention to green fashion, including green clothing design and fabric, consumption, vintage issues, knowledge-sharing about manufacturing processes, materials and fabrics, reusing and recycling of fashion products (vintage), transportation, and distribution. [42]. Xu et al. identify the impact of the social, economic, and environmental dimensions of sustainability in the hospitality industry, especially the supply chain activities on consumer behaviors and attitudes, including loyalty, satisfaction, and readiness to pay a premium for sustainable services and goods. The results of the study confirm that customer satisfaction is enhanced by environmentally friendly actions [43]. In contrast, more developed aspects are related to energy and material consumption and the emissions. In the social indicators,

workers and local community-based aspects are given more attention as compared to the consumer-based aspects. [44].

Understanding and introducing certain sustainability practices, namely, environmental, social, and economic aspects, at the supply chain interface allows firms to identify the performance and importance of various supply chain sustainability practices to achieve and maintain a long-term competitive advantage. The findings of the study reveal that the hotel industry managers give the most significant attention to sustainability practices within the firm's operations. The managers' perception of sustainable practices in the entire supply chain varies according to the members of the supply chain [45,46]. Lu et al. proposed that once a company has clear sustainability practices, policies, and participation, it is more likely to cooperate with partners, customers, and external suppliers. The firm will be better able to understand its business, which will improve and facilitate collaboration within the supply chain. Therefore, the study confirms that firms that focus on sustainability practices will have improved collaboration with their customers and suppliers [47]. Gouda et al. proposed research for the risk identification and evaluation related to sustainable supply chains. They expanded their survey to 21 different countries and gathered the findings for the risk-mitigating actions related to sustainable supply chains. They found that the risk mitigation actions are usually reactive rather than proactive and continuous efforts are needed to address these risks [48]. Harms et al. addressed the issue of selecting sustainable suppliers for the large firms in Germany. They adopted the strategic methodology and acquired the opinions of the experts using the survey. They concluded that the supplier evaluation and selection requirements are dynamic and hence, add risks to the system. Hence, it is necessary to monitor and align those indicators [49] continuously. Hong et al. presented a dynamic model including the changing requirements and their effects on the sustainable supply chains. They found the dynamic capabilities more convenient and effective [50]. Colicchia et al. investigated the compliance level of multi-national organizations for the sustainable supply chain framework. They developed a unique framework for the sustainable supply chain, incorporating most of the factors/indicators, implemented the framework, and compared the organization having the framework implemented and those that are not [39]. Elmuti et al. developed a linkage between the internal factors of a sustainable supply chain and the external factors for sustainability compliance. They found that rather than complying only with the internal supply chain factors, compliance with the external factors make organizations more responsive toward compatibility and acquiring new suppliers [51]. Table 1 illustrates a summarized form of the literature.

**Table 1.** Literature review addressing the green supply chain and sustainability issues.

| Author(s) | Sustainability(General) | Environmental Factors | Social Factor | Economic Factor | Other/Focus |
|---|---|---|---|---|---|
| Khan et al. [2] | ✓ | ✓ | ✓ | ✓ | South Asia |
| Feng et al. [3] | ✓ | | | ✓ | |
| Yun et al. [8] | ✓ | ✓ | | ✓ | |
| Khan et al. [10] | ✓ | ✓ | ✓ | ✓ | Quality |
| Longoni et al. [11] | ✓ | ✓ | | | |
| Adomako et al. [12] | ✓ | ✓ | | | Green HR |
| Zaid et al. [13] | ✓ | ✓ | | | Green HR |
| Baah et al. [14] | ✓ | ✓ | | | |
| Green et al. [15] | ✓ | | | | JIT, TQM |
| Carter et al. [16] | ✓ | ✓ | ✓ | ✓ | |
| Montabon et al. [17] | ✓ | | ✓ | | |
| Markman et al. [18] | ✓ | | ✓ | | |
| Karmaker et al. [19] | ✓ | | | ✓ | Finances |
| HE et al. [20] | ✓ | | | | Knowledge Management |
| Radi et al. [21] | ✓ | ✓ | ✓ | ✓ | Social media analytics |
| Dong et al. [22] | ✓ | | | | Behavioral aspect |
| Kusi-Sarpong [23] | ✓ | ✓ | ✓ | ✓ | Innovation Mgmt. |
| Walker et al. [24] | ✓ | | | ✓ | UK |
| Martins et al. [25] | ✓ | ✓ | ✓ | | Brazillion |
| Ghoushchi et al. [26] | ✓ | ✓ | ✓ | ✓ | GP-DAE |
| Chabra et al. [27] | ✓ | ✓ | ✓ | ✓ | Indian |
| Halkos et al. [28] | ✓ | | ✓ | | CSR |

**Table 1.** *Cont.*

| Author(s) | Sustainability(General) | Environmental Factors | Social Factor | Economic Factor | Other/Focus |
|---|---|---|---|---|---|
| Chu et al. [29] | ✓ | | | | Organizational culture |
| La Scalia et al. [30] | ✓ | ✓ | | | Ecological impact |
| Plambeck et al. [31] | ✓ | ✓ | | | Clean energy |
| Khan et al. [32] | ✓ | ✓ | | | Clean energy |
| Helo et al. [33] | ✓ | ✓ | | | Clean energy |
| Hojnik et al. [34] | ✓ | ✓ | ✓ | ✓ | Yachting Industry |
| Stindt et al. [35] | ✓ | ✓ | | | Planning |
| Vasileiou et al. [36] | ✓ | ✓ | | | Potatoes, UK |
| Zailani et al. [37] | ✓ | | | | Malaysia |
| Kalenoja et al. [38] | ✓ | ✓ | | | Clean energy |
| Colicchia et al. [39] | ✓ | ✓ | ✓ | ✓ | Benchmarking |
| Kim et al. [40] | ✓ | ✓ | ✓ | ✓ | |
| Modica et al. [41] | ✓ | | | | Hospital Industry |
| Cervellon et al. [42] | ✓ | | | | Fashion Industry |
| Xu et al. [43] | | | ✓ | | Attitude & Behavior |
| Ahmed et al. [45] | ✓ | ✓ | ✓ | ✓ | Manufacturing Industry |
| Babu et al. [46] | ✓ | ✓ | ✓ | ✓ | Hotel Industry |
| Lu et al. [47] | ✓ | ✓ | ✓ | ✓ | Container Terminal |
| Gouda et al. [48] | ✓ | ✓ | ✓ | | OECD countries |
| Harms et al. [49] | ✓ | ✓ | ✓ | ✓ | German Supplier Dev. |
| Hong et al. [50] | ✓ | ✓ | ✓ | ✓ | Chinese Manufacturers |
| Elumti et al. [51] | ✓ | | | | Compatibility |

■ After conducting a comprehensive literature review, immense research gaps have been identified. A few studies are present on the supply chain professionals' perception for the sustainability indicators. While adopting sustainable supply chain practices, the professionals give importance to the core indicators of sustainability, primarily environmental and economic, negating the other sustainability aspects to the second level and paying less attention to these [25]. However, studies suggest that it is imperative to consider all three sustainability indicators for optimum supply chain and organizational performance equally [26]. Therefore, the proposed research is defined based on the consideration to all the three sustainability indicators equally. While reviewing the literature, it is found that supply chain professionals pay less importance to social indicators while thinking about sustainable supply chain processes. Therefore, to improve the performance of a sustainable supply chain, it is also essential to assess which sustainability indicator is perceived more critical for others by the supply chain professionals. Moreover, it is also essential to analyze the correlation between the indicators for improved supply chain performance. In the stated literature review, the supply chain's sustainability is usually assessed for the specific industry and specific area of interest, which is also the limitation of the previous research. Based on these gaps, the following objectives are set for the current research: To assess the opinions of professionals from the diversified industries to analyze the sustainability of the supply chain.

■ Based on the opinions of a supply chain professionals, determine the relative importance of sustainability indicators, namely environmental, social, and economic.

■ To determine the relationship among any of these sustainability indicators, to improve the supply chain performance.

■ To assess the importance of sustainability indicators with for the variation in gender.

To attain the defined objectives, a set of hypotheses have been built, which are as follows:

**Hypothesis 1 (H1).** *Supply chain practitioners attribute more importance to all three environmental, social, and economic indicators.*

**Hypothesis 2 (H2).** *The perception of supply chain practitioners about environmental, social, and economic impacts are influenced by gender.*

**Hypothesis 3 (H3).** *The experience of supply chain practitioners impacts the importance of environmental, social, and economic indicators.*

**Hypothesis 4 (H4).** *Environmental indicators impact the economic indicators.*

**Hypothesis 5 (H5).** *Social indicators impact the environmental indicators.*

**Hypothesis 6 (H6).** *Environmental indicators impact through the mediation of social indicators to the economic factors.*

Later to this introduction section, there are four more sections. Section 2 describes the methodological procedures used in similar studies. A quantitative research method is adopted. Different techniques are applied, such as regression analysis, independent sample t-test, cluster analysis, Cronbach Alpha, factor loading, and structural equation modelling (SEM) for analyzing the data. Section 3 addresses the survey findings, and Section 4 will end the article with the conclusion, recommendations, and future research direction.

## 2. Methodology

To test the hypotheses mentioned above, an investigation on the significance of supply chain professionals' perception has been performed. It is applied to all three indicators with the help of 33 sub-items developed under all three sustainability indicators; environmental, social, and economic (see Figure 1).

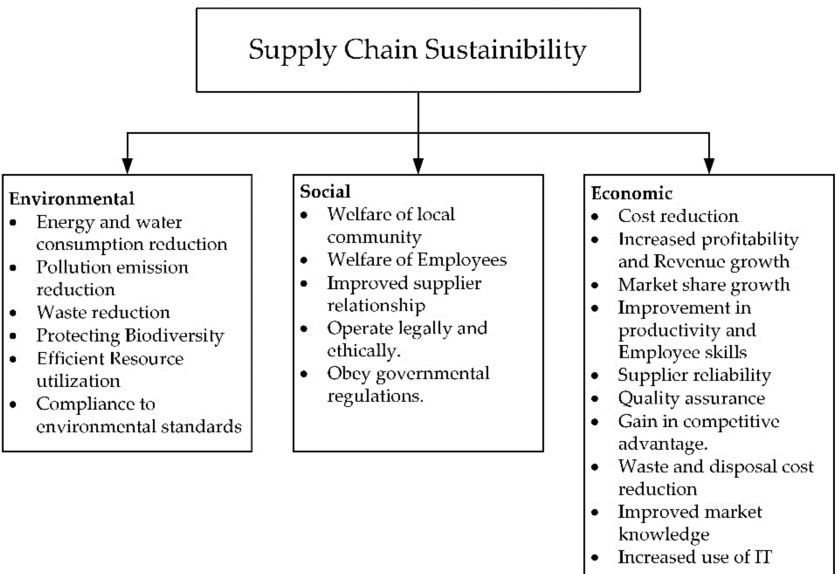

**Figure 1.** Dimensions of supply chain sustainability.

The flow of methodology is shown in Figure 2.

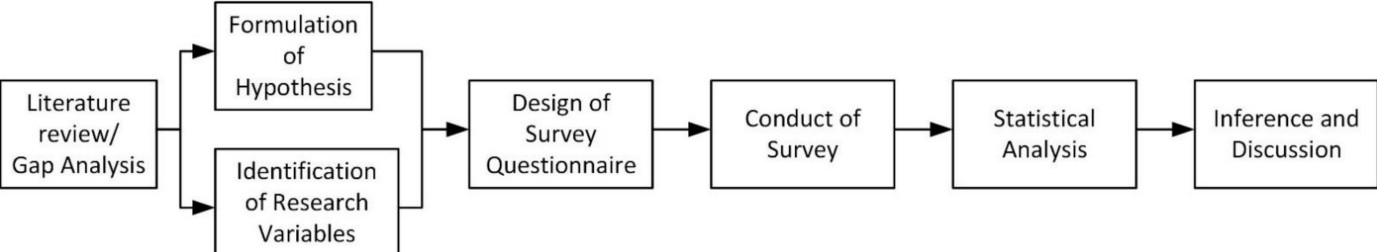

**Figure 2.** Research Methodology.

A review of previous literature is conducted to understand the supply chain sustainability and implementation of sustainability into the three indicators. A gap is identified in the literature that shows that the analysis of the perception of supply chain professionals on sustainability is an important aspect. It questions how much importance do they give to all the indicators of sustainability?

A research framework is constructed along with the identified variables that will be used to investigate the validity of the hypotheses in the research. A construct is built on "sustainability" based on three indicators, namely environmental, social, and economic. A research questionnaire is developed based on the variables identified with the help of the literature review for primary data collection to empirically test the perceptions of supply chain professionals about the sustainable supply chain. A five-point Likert scale ranging from Not Important equal to one to Essential equal to five is selected for the questionnaire. The questionnaire consists of 2 sections; the first section consists of general information required to identify the profile of the respondents. The second section includes items related to three sustainability indicators. All items of the questionnaire are adopted from the literature review (see Table 2). The reliability (Cronbach Alpha) and validity (CFA or EFA or Factor Loading) of this instrument are also assessed to finalize the questionnaire.

**Table 2.** Questionnaire Items.

| Items | Reference |
|---|---|
| **Sustainability Indicator: Environmental (Enviro #)** | |
| 1. Environmental certifications (e.g., EMAS or ISO 14001)<br>2. Energy and water consumption reduction program<br>3. Pollution emission reduction and waste recycling program | [48] |
| 4. Waste reduction<br>5. Renewable energy usage<br>6. Protecting Biodiversity | [49] |
| 7. Efficient Resource utilization | [50] |
| 8. Using environmental management systems | [35] |
| 9. Compliance to environmental standards | [37] |
| 10. Green purchasing | [41] |
| 11. Selecting green modes of transportation<br>12. Warehousing should include using green sources of energy and green building practices | [51] |
| **Sustainability Indicator: Social (Socio #)** | |
| 1. Social certifications (e.g., SA8000 or OHSAS 18000)<br>2. Work/life balance policies | [48] |
| 3. Health protection of community members<br>4. Ensuring Human rights<br>5. Child and forced labour avoidance<br>6. Job security for employees | [49] |
| 7. Employee perspective considered | [51] |
| 8. Engage in employment diversity<br>9. Promote fair treatment of all employees<br>10. Create a safe and healthy work environment<br>11. Provide training and development for employees<br>12. Encourage employee participation in community projects<br>13. Provide financial support for community activities<br>14. Establish long-term partnerships with suppliers<br>15. Operate legally and ethically<br>16. Create partnerships with government agencies | [41] |

**Table 2.** *Cont.*

| Items | Reference |
|---|---|
| **Sustainability Indicator: Economic (Econom #)** | |
| 1. Supplier reliability<br>2. Quality assurance<br>3. Cost reduction<br>4. Innovation potential | [49] |
| 5. Reduce cost, inventory, and cycle time<br>6. Improve delivery and reliability—customer service<br>7. Increase productivity<br>8. Increase market share<br>9. Focus on core competencies<br>10. Gain competitive advantage | [51] |
| 11. Improved overall profitability and revenue growth.<br>12. Improved market knowledge<br>13. Improved employee skills<br>14. Increased use of IT | [41] |

The survey was conducted based on an individual sample unit as the survey was sent to the supply chain professionals. Contacts were gathered earlier, to survey the convenience sampling technique as the sample involved supply chain professionals at different levels, such as managers, directors, or supervisors of different industries. The questionnaire was sent to 1000 plus practitioners, out of which 106 responses were received, but due to missing data, one response was excluded. Thus, the sample size consisted of 105 practitioners.

Analysis of data was performed later, after the completion of the online survey. The responses from the survey are analyzed using the SPSS and STATA software (IBM, Armonk, NY, USA). For data analysis, regression analysis, independent sample t-test, and cluster analysis are applied. For reliability and validity assessment, content validity index (CVI) and Cronbach Alpha and factor loading methods are applied, respectively. All these analyses are applied using the IBM SPSS software (IBM, Armonk, NY, USA). Structured equation modelling (SEM) technique is also applied using the STATA as research conducted in these domains also used this statistical analysis tool [18,52]. Hence, in this article, various statistical techniques are applied for the analysis of data and assessing the validity of the hypotheses.

## 3. Analysis

The statistics of respondents showed that 25% of responses were from the service industry, 21% responses were from the manufacturing industry, and 18% were from the FMCG industry. The remaining 10% of responses were from the energy sector, 4% government and the remaining from other sectors. The respondents' profile also showed that 68% of respondents had up to 5 years of supply chain experience, 20% of respondents had 6 to 11 years' experience in the supply chain, and 10% of respondents had 11 to 15 years of supply chain experience, and the remaining 3% had around 20 years of experience. The respondents' designations showed that 69% were assistant managers to manager or equivalents, 10% were deputy general managers or general managers or equivalents, and 3% were directors or equivalent. The organization size showed that 51% of respondents belong to organizations that had more than 500 employees, 21% had up to 200 employees, and 14% had up to 500 employees. All these statistics showed that respondent profiles meet the criteria we required for the survey.

### 3.1. Content Validation

The questionnaire is reviewed by the three highly experienced researchers having PhD degrees in supply chain management. The content validity of the questionnaire is also determined using the opinions of the three experts. The questionnaire is finalized after

a content validity index (S-CVI/Ave) is calculated to be 0.845, which is an acceptable index for the survey questionnaire (more than 0.8).

### 3.2. Reliability of Questionnaire

For the reliability assessment, Cronbach Alpha is used. It showed that internal consistency exists with a Cronbach alpha value of 0.962, which is above the satisfaction level of 0.7.

### 3.3. Factor Loading

For validity, the factor loadings method is applied (see Table 3). The coefficient must be more than 0.5 for all the factors, which shows the construct validity. It also showed that variable Enviro7 for environmental indicator, Socio5, Socio14, 15, 16 for social indicators and Econom1 for economic indicators were below 0.5. Hence, these variables are omitted for further analysis.

**Table 3.** Factor Loadings.

| Variables | Factor Loadings | Variables | Factor Loadings | Variables | Factor Loadings |
|---|---|---|---|---|---|
| Enviro1 | 0.859 | Socio1 | 0.686 | Econom1 | |
| Enviro2 | 0.83 | Socio2 | 0.577 | Econom2 | 0.638 |
| Enviro3 | 0.875 | Socio3 | 0.605 | Econom3 | 0.689 |
| Enviro4 | 0.842 | Socio4 | 0.669 | Econom4 | 0.807 |
| Enviro5 | 0.726 | Socio5 | | Econom5 | 0.641 |
| Enviro6 | 0.858 | Socio6 | 0.525 | Econom6 | 0.681 |
| Enviro7 | | Socio7 | 0.617 | Econom7 | 0.743 |
| Enviro8 | 0.868 | Socio8 | 0.579 | Econom8 | 0.754 |
| Enviro9 | 0.87 | Socio9 | 0.773 | Econom9 | 0.771 |
| Enviro10 | 0.851 | Socio10 | 0.757 | Econom10 | 0.826 |
| Enviro11 | 0.667 | Socio11 | 0.573 | Econom11 | 0.746 |
| Enviro12 | 0.782 | Socio12 | 0.652 | Econom12 | 0.716 |
| | | Socio13 | 0.584 | Econom13 | 0.682 |
| | | Socio14 | | Econom14 | 0.583 |
| | | Socio15 | | | |
| | | Socio16 | | | |

### 3.4. Cluster Analysis

Through cluster analysis, it is found that H1 is statistically supported (see Table 4). The results of cluster analysis show that there are three types of groups in our sample. The first group is the one that attributes less importance to all of the three indicators. Then, there is a second group that attributes medium importance to all of the three indicators, and then the third group attributes high importance to all of the three indicators. The summary of results shows that cluster three has 69 people who attribute importance to all three indicators, almost 66% of our sample. The one that falls under cluster one is an outlier. Additionally, through the ANOVA results, we found a significant variation in each item of all three indicators selected for the survey.

### 3.5. Independent Sample t-Test

Table 5 presents the group statistics for the sustainability indicators. It shows that more respondents, either male or female, agreed on the significance of social and economic indicators, whereas they attribute low significance to the environmental indicator. To assess the validation of the hypotheses H2 and H3, independent sample *t*-tests are applied (see Tables 5 and 6). It was found that the gender of the supply chain practitioners does not

impact perception about the social and economic indicators. However, there is a significant difference ($p > 0.05$) between male and female practitioners for the environmental indicators. The mean of males (42.2) is higher than the females (36.1), which shows that males perceive environmental indicators to be more important than females.

**Table 4.** Number of Cases in each Cluster.

| | | |
|---|:---:|:---:|
| | 1 | 1 |
| **Cluster** | 2 | 35 |
| | 3 | 69 |
| **Valid** | | 105 |
| **Missing** | | 1 |

**Table 5.** Group Statistics.

| | Gender | N | Mean | Std. Deviation | Std. Error Mean |
|---|:---:|:---:|:---:|:---:|:---:|
| Environmental Indicator | Male | 67 | 42.2090 | 12.05359 | 1.47258 |
| | Female | 38 | 36.1053 | 13.45621 | 2.18289 |
| Social Indicator | Male | 67 | 61.8507 | 11.12226 | 1.35880 |
| | Female | 38 | 60.9211 | 8.99364 | 1.45896 |
| Economic Indicator | Male | 67 | 57.0000 | 10.11450 | 1.23568 |
| | Female | 38 | 60.2105 | 7.38939 | 1.19872 |

**Table 6.** Independent Samples Test.

| | | Levene's Test for Equality of Variances | | t-Test for Equality of Means | | | | | | |
|---|---|:---:|:---:|:---:|:---:|:---:|:---:|:---:|:---:|:---:|
| | | F | Sig. | T | df | Sig. (2-tailed) | Mean Diff. | Std. Error Diff. | 95% Confidence Interval of the Difference | |
| | | | | | | | | | Lower | Upper |
| Environmental Indicator | Equal variances assumed | 1.706 | 0.194 | 2.390 | 103 | 0.019 | 6.103 | 2.553 | 1.038 | 11.168 |
| | Equal variances not assumed | | | 2.318 | 70.190 | 0.023 | 6.103 | 2.633 | 0.852 | 11.355 |
| SocialIndicator | Equal variances assumed | 0.707 | 0.402 | 0.440 | 103 | 0.661 | 0.929 | 2.113 | -3.262 | 5.121 |
| | Equal variances not assumed | | | 0.466 | 90.749 | 0.642 | 0.929 | 1.993 | -3.030 | 4.890 |
| EconomicIndicator | Equal variances assumed | 2.087 | 0.152 | −1.713 | 103 | 0.090 | −3.210 | 1.874 | −6.927 | 0.5064 |
| | Equal variances not assumed | | | −1.865 | 96.394 | 0.065 | −3.210 | 1.721 | −6.627 | 0.2066 |

To check if there is any difference in the responses based on experience, an independent sample t-test was run again. It was found that the experience of supply chain practitioners does not impact perception about the environmental, social, and economic indicators.

### 3.6. Regression Analysis

Through regression (see Model 1 in Table 7), it is found that the hypothesis H4 is statistically supported. Economic indicators impact the environmental indicator as the adjusted R Square is 0.117, which means almost a 12% perception impacts the economic indicators. The value of beta (0.355) shows a weak positive relation. Additionally, through ANOVA (see Table 8), it is found that the model is statistically significant ($p \leq 0.05$). Therefore, the effect of environmental indicators on economic indicators has a weak relation.

For the hypothesis H5, regression analysis is applied, as shown in Table 7 (model 2). It also shows that the hypothesis is statistically supported. The environmental indicator is impacted moderately by the social indicators as the adjusted R Square is 0.426, which means almost a 43% impact over the environmental indicators. Additionally, through

ANOVA (see Table 9), it is found that the model is statistically significant ($p \leq 0.05$). The value of beta (0.657) shows a moderate positive relationship.

**Table 7.** Regression Analysis.

| Model | R | $R^{2.5}$ | Adjusted $R^2$ | Std. Error of the Estimate | Change Statistics | | | | |
|---|---|---|---|---|---|---|---|---|---|
| | | | | | $R^2$ Change | F Change | df1 | df2 | Sig. F Change |
| 1 | 0.355 [a] | 0.126 | 0.117 | 0.62500 | 0.126 | 14.842 | 1 | 103 | 0.000 |
| 2 | 0.657 [b] | 0.432 | 0.426 | 0.81158 | 0.432 | 78.261 | 1 | 103 | 0.000 |

[a]. Predictors: (Constant), EnviroIndicator1; [b]. Predictors: (Constant), SocialIndicator1.

**Table 8.** Analysis of Variance (ANOVA) with respect to Economic Indicator.

| Model | | Sum of Squares | Df | Mean Square | F | Sig. |
|---|---|---|---|---|---|---|
| | Regression | 5.798 | 1 | 5.798 | 14.842 | 0.000 [a] |
| 1 | Residual | 40.234 | 103 | 0.391 | | |
| | Total | 46.032 | 104 | | | |

[a]. Predictors: (Constant), EnviroIndicator1.

**Table 9.** Analysis of Variance (ANOVA) for Economic Indicator.

| Model | | Sum of Squares | Df | Mean Square | F | Sig. |
|---|---|---|---|---|---|---|
| | Regression | 51.547 | 1 | 51.547 | 78.261 | 0.000 [a] |
| 1 | Residual | 67.842 | 103 | 0.659 | | |
| | Total | 119.389 | 104 | | | |

[a]. Predictors: (Constant), SocialIndicator1.

### 3.7. Structural Equation Modeling

The result of structural equation modelling suggests that the hypothesis H6 is not statistically supported. Hence, there is no effect of mediation of social indicators. It is found in the results that social indicators do not mediate the relation between environmental and economic indicators. SEM results show no goodness of fit of this model as the value of chi-square is significant at 0.00. Additionally, the value of RMSEA (0.112) is greater than 0.08, which validates that the model is not a good fit. The CFI and TLI values are 0.722 and 0.707, which are not appropriate as these should be greater than 0.9 for a model to be a good fit.

### 4. Discussion

To compete in the era of globalization and technological advancement, it has become crucial for the supply chain management to improve process productivity. Companies must conduct their operations following the triple bottom line rule, to be sustainable through the implementation of green supply chain practices. Despite the high costs of the green supply chain and to be competitive, firms need to integrate sustainable practices in their supply chain activities as per customer demands and environmental regulations [29].

The majority of the sustainable practices adopted for companies in supply chain systems aim towards the economic aspect to reduce operational costs and give little attention to sustainability's environmental and social aspects. However, to improve supply chain and organizational performance, companies should focus on implementing all sustainability indicators, which is evident from the results that support the trueness of Hypothesis 1 (H1). Thus, the environmental, social, and economic aspects of sustainability need to be considered by the firm when reengineering or designing supply chain operations [30]. To improve the performance of a sustainable supply chain, it is also important to assess which sustainability indicators, namely environmental, social, and economic, are perceived as more important for others, by supply chain professionals.

This study investigates the perception of supply chain practitioners towards the following three dimensions of sustainability: environmental, social, and economic indicators. By investigating the relative importance of these three dimensions of sustainability, this study makes important theoretical contributions to the field of supply chain management. Contrary to previous studies that suggested that supply chain practitioners give importance to only one or two dimensions of sustainability, considering the social indicators of sustainability in logistics as secondary, [25] the findings reported in this study suggest that all three dimensions of sustainability (environmental, social, and economic) are perceived as equally important by the majority of supply chain practitioners when it comes to the implementation of green supply chain practices. These findings are in line with the findings of another study, which concludes that the firm's environmental, social, and economic aspects need to be considered by the firm when reengineering or designing sustainable supply chain operations and improved decision-making and business performance [30,36]. These findings are also aligned with the results of research conducted during the COVID-19 pandemic, which proposes that for a sustainable supply chain to be effective, sustainable strategies are important and these strategies should include factors from all three indicators of the TBL [19] Additionally, it is consistent with the results of the findings of He et al., which indicate that practicing managers' perception supports strategies that are found from the lens of the TBL [20]. Though all the factors of the three sustainability dimensions are important for sustainable supply chain management, this study also suggests that there are some factors in each dimension that are perceived to be relatively more important than other factors by the practitioners. This finding is in line with one study, which emphasizes that within the environmental, social, and economic dimension of sustainability, the relative importance of sub-indicators varies from a triple bottom line perspective [20].

The findings in this research also suggest that the perception of the relative importance of these three dimensions of supply chain sustainability does not differ from the experience level of supply chain practitioners (Hypothesis 3—H3), which is in contrast to the finding that managers' perception of sustainable practices in the entire supply chain varies according to the characteristics of the members of the supply chain [46]. The findings of this study also show that even though the perception of the relative importance of social and environmental dimensions is not significantly impacted by the gender (Hypothesis 2—H2) of the supply chain practitioners, the perception of environmental factors does vary with gender. The results of this study suggest that male practitioners perceive environmental factors as more important for sustainable supply chain implementation than female practitioners. Both these findings are consistent with the results of [46], that the managers' perception of sustainable practices in the entire supply chain varies according to the characteristics of the members of the supply chain [46].

According to the findings, in the environmental dimension, the top three factors perceived as highly important are compliant with the environmental standards, pollution emission reduction and waste recycling program, and using environmental management systems. These results are consistent with a previous study that showed that solid waste, waste water, and water usage are the less developed environmental indicators. In contrast, more developed aspects are related to energy and material usage and air emissions [45]. These findings are also consistent with the results of a study that shows that waste and pollution reduction from a consumer and eco-friendly systems are the most important factors of sustainability within the environmental dimension [21]. However, the findings in this research are in contrast to the results of a previous study that shows that instead of all three dimensions, only the environmental dimension is considered most important by the practitioner. However, a gap exists between importance and performance, as only giving importance to the environment does not ensure sustainable supply chain performance [46]. The results also shows that the environmental dimension of the sustainability also affects the economic indicators (Hypotheses 4—H4). It is due to the fact that the environmental compliance incurs capital, the cost of operations, and especially the disposal cost, which directly affects the economy of the projects.

In the social dimension, the highest importance is given to encouraging employee participation in community projects and then to the health protection of the community members. This finding is in line with the results, which indicates that aspects of social indicators, such as workers and local community-based aspects, are given more attention than consumer-based aspects [45]. It is also in line with the findings within the social dimension that human rights are most important. The research supports the Hypotheses 5—H5 that the social dimension has impact over the environmental indicators. Since the encouragement of employee participation in community projects and the health protection community members are the factors that directly affect the environment indicators. The involvement, training, and participation of employees encourage them to be more responsive towards the environment, resulting in the positive impact of these over the environmental indicators.

The results also highlighted that in the economic dimension, supplier reliability, quality assurance, and reducing cost, cycle time and inventory are given the highest importance, respectively, which is in line with the results of a past study showing that from an economic viewpoint, more importance is given to the cost-based aspects than the profit-based indicators [45]. The result of this study shows that all three dimensions of sustainability are perceived to be equally important by supply chain practitioners, which is contradictory to the result of the study that showed that the environmental dimension is perceived as the most important [21].

The findings of this study also show that the environmental dimensions of sustainability on the economic dimensions have a weak but positive relationship, suggesting that the perception of environmental factors weakly impacts the perception of economic indicators. These findings are in line with the results of research and that show that environmental sustainability indicators are directly related to the economic objectives. To achieve sustainable targets in the supply chain, it is also important to identify a correlation between the sustainability indicators of the supply chain [25].

However, the relationship between the impact of social dimension on the economic dimension of sustainability has a moderately positive relationship (Hypothesis 6—H6), suggesting that the perceived importance of the social dimension of supply chain practitioners moderately impacts their perceived importance of economic indicators. This study identifies that social indicators do not mediate the relationship between the perceived importance of environmental and economic indicators. Considering the emphasis of Golroudbary et al. research for the better implementation of sustainability in the supply chain, it is necessary to promote, develop, and recognize sustainable policies and practices to ensure that there is a fair balance between economic, environmental, and social elements [53]. This study adds to the extant literature by contributing that all of the three dimensions of sustainability, namely, economic, environmental, and social, need to be considered equally important by supply chain practitioners for the successful implementation of sustainable supply chain management. These findings are consistent with the emphases of Dong et al., who say that it is critical to give importance to behavioral concerns or perceptions, as they significantly impact efficiency and effectiveness in the sustainable supply chain [22]. However, this is in opposition to the findings of previous research, which emphasizes that the economic dimension is the most important and critical to achieving innovative sustainable supply chain [23,54].

## 5. Conclusions

This study applies statistical analysis techniques to investigate the relative importance of sustainability indicators, namely economic, environmental, and social, as perceived by supply chain practitioners to implement sustainable supply chain practices. The results indicate that supply chain practitioners perceive all three dimensions of sustainability as equally important. A positive relationship exists between the impact of environmental indicators on economic indicators and between the impact of social indicators on economic indicators. Lastly, the perception of environmental indicators varies with gender,

as male practitioners perceive these indicators as more important than females. The findings suggest that supply chain practitioners give importance to and hence adopt all three dimensions of sustainability, economic, environmental, and social, for the efficient and successful implementation of sustainable supply chains.

The research suggests that compliance to environmental standards, pollution emission reduction and waste recycling programs are the most important environmental factors in sustainable supply chains. It also suggests that supplier reliability, quality assurance, reducing cost, cycle time, and inventory are important factors in the economic category, whereas encouraging employee participation in community projects and the health protection of community members are the weighted factors in the social category. Hence, controlling these factors may result in a sustainable supply chain in either manufacturing or other industries.

This study makes meaningful practical and theoretical contributions to sustainable supply chain management; however, it is not without limitations. One of the limitations of our study is that a few variables are not included in the research model for the sake of parsimony, such as product quality, product design, product price, perceived risk, and organizational culture. Secondly, as this study is based on survey design, some subjective measures might suffer from method bias, such as social desirability and item ambiguity.

Future research on sustainable supply chain management can study the three sustainability dimensions on performance or productivity. Variables such as product quality, product design, product price, perceived risk, and organizational culture should also be included in future research to check their impact on supply chain sustainability. To improve the generalizability, future research can replicate the study by including a greater number of larger companies, 3PL providers, and logistics firms from different countries to avoid the noise caused by countries and firm size. The framework of the sustainability dimensions in this study can also be tested in other geographical areas. Researchers can test the sustainability framework by finding out the impact of antecedent variables on other measures of performance and productivity in future research.

**Author Contributions:** Conceptualization, M.W. and R.K.; methodology, M.W., S.H., S.K.; software, S.H. and S.K.; validation, R.K., M.W., S.H. All authors have read and agreed to the published version of the manuscript.

**Funding:** This research received no external funding.

**Institutional Review Board Statement:** Not applicable.

**Informed Consent Statement:** Not applicable.

**Conflicts of Interest:** There is no conflict of interest related to this article and the authors.

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
