# Peer review of "Supply Chain Practitioners’ Perception on Sustainability: An Empirical Study"

_sustainability, doi:10.3390/su13179872_

Round 1
Reviewer 1 Report
In this paper, the authors have studied supply chain practitioners' perceptions on sustainability. However, some of my concerns regarding this paper are as follows:
- There is a lack of novelty. The authors are advised to describe research gaps and their contributions in elucidating manner.
- A table must be included to show the literature gaps and what are new factors or methodologies have adopted in the current study.
- Figures quality need to be improved. In some figures, the arrows are not in the correct place.
- The section structure should be modified (i.e. The literature review section should be after introduction.... etc..)
- The managerial implication should be described on the basis of the result obtained.
- Future scope and limitation of the proposed work is missing.
Author Response
We are thankful to you for considering our manuscript for your esteemed journal and also thankful to the reviewer who contributed to the improvement of our article.
Reviewer #1
In this paper, the authors have studied supply chain practitioners' perceptions on sustainability. However, some of my concerns regarding this paper are as follows:
|
Comment |
Correction |
|
1.There is a lack of novelty. The authors are advised to describe research gaps and their contributions in elucidating manner. |
Research novelty and contribution is re-written in a more elaborative way. Please view page 4, after the table. I have highlighted it in yellow colour. |
|
2. A table must be included to show the literature gaps and what are new factors or methodologies have adopted in the current study.
|
Table 1 has been inserted to show the research gap. This table is on pages 3 and 4. The title of the table is highlighted in yellow colour. |
|
3. Figures quality need to be improved. In some figures, the arrows are not in the correct place.
|
Fig. 2 (page 6) and Fig. 3 (page 7) are revised with high quality. |
|
4. The section structure should be modified (i.e. The literature review section should be after introduction.... etc. |
The introduction section is entirely revised according to the reviewer's suggestion. Please refer to pages 1 to 4. |
|
5. The managerial implication should be described on the basis of the result obtained. |
Managerial implications based on the results are highlighted in yellow colour in the Discussion section on pages 12 and 13. |
|
6. Future scope and limitation of the proposed work is missing |
Limitation and future of scope are separately mentioned on page 14 and are highlighted in yellow color. |

Reviewer 2 Report
The English level should be enhanced.
The structure of the Conclusions must be modified.
Author Response
We are thankful to you for considering our manuscript for your esteemed journal and thankful to the reviewer who contributed to our article's improvement.
Reviewer #2
The authors have carried out number of experiments to understand the spring back effect. The data are useful. They have cited lot of relevant references as well. However, in my opinion few points have to be explained properly.
|
Comment |
Correction |
|
1. The English level should be enhanced.
|
The article is critically reviewed and the language is improved as per the reviewer's suggestions. Almost all sections are revised. |
|
2. The structure of the Conclusions must be modified. |
The conclusion (page 14) section is modified. More details are inserted, which is highlighted in green colour. Future Scope and limitation of work are separated from the conclusion section. |

Reviewer 3 Report
This study uses SEM to analyze the perception of practitioners on supply chain sustainability. I do not see any contributions to this work. My major concerns are detailed below.
- The background/introductory part of the abstract is very weak. The abstract should be shortened to less than 180 words; you included way too many details.
- Some of the keywords are redundant.
- The introduction section should be completely restructured. You do not need to have sub-sections; instead, the following paragraph should be provided: background, problem statement, the most relevant research works that addressed the problem (with a critical view), a clear statement of the gap, contribution(s), and research gaps, and outlines of your manuscript.
- There are many published works on this very subject. (1) you have not reviewed many relevant works. (2) How your work is different from theirs?
- The literature review is VERY selective. You did not follow a systematic way to review the relevant literature.
- The method is not well explained. It should be so clear that potential readers can replicate it in their research. Even though the Structural Equation Modeling literature is abundant, the manuscript should be self-sufficient.
- The conclusion is not supported by the findings.
Author Response
We are thankful to you for considering our manuscript for your esteemed journal and thankful to the reviewer who contributed to our article's improvement.
Reviewer #3
This study uses SEM to analyze the perception of practitioners on supply chain sustainability. I do not see any contributions to this work. My major concerns are detailed below.
|
Comment |
Correction |
|
1. The background/introductory part of the abstract is very weak. The abstract should be shortened to less than 180 words; you included way too many details. |
The abstract has been revised with the change in the introductory part. It is not limited to 180 words as per the reviewer's suggestions. |
|
2. Some of the keywords are redundant. |
Redundant keywords are eliminated, new keywords are added as per the reviewer's suggestions. |
|
3. The introduction section should be completely restructured. You do not need to have sub-sections; instead, the following paragraph should be provided: background, problem statement, the most relevant research works that addressed the problem (with a critical view), a clear statement of the gap, contribution(s), and research gaps, and outlines of your manuscript. |
The introduction section is revised according to the suggestions. The section discusses the fundamentals of the supply chain, green supply chain, sustainability. Then it discusses a detailed literature review. The research gap, research problem and novelty is discussed. Table 1 is added to include a summary view of the literature review, which is easy to extract the research gap. Research objectives are now clearly described in the introduction section (see page 1 to 4). |
|
4. There are many published works on this very subject. (1) you have not reviewed many relevant works. (2) How your work is different from theirs? |
Few more literature review detail has been added, which was previously only cited but the review was not inserted. It is inserted on pages 3 and 4 and is highlighted in cyan colour. Just after Table 1, the research gap, problem and novely is discussed in detail, highlighted in yellow (see page 5). It presents the difference between the current literature review and the contribution in the field. |
|
5. The literature review is VERY selective. You did not follow a systematic way to review the relevant literature. |
A total of 56 research articles are cited, with a literature review of 48 research articles. It is requested that the reviewers consider and accept this literature review as it is detailed and sectioned as per the topic of the discussion. |
|
6. The method is not well explained. It should be so clear that potential readers can replicate it in their research. Even though the Structural Equation Modeling literature is abundant, the manuscript should be self-sufficient. |
The structure of Section 2 (Methodology) is revised. Now the hypothesis is also included in the same section. Figure 2 is improved, which is explained in detail. (see page 5 to 8) |
|
7. The conclusion is not supported by the findings. |
The conclusion (page 14) section is modified. More details are inserted to support the findings, which is highlighted in green colour. Future Scope and limitation of work are separated from the conclusion section. |

Round 2
Reviewer 1 Report
The authors have addressed all the comments and suggestions in a satisfactory manner. Thus, I recommend it's for publication.
Author Response
Thank you very much for the review of our article.
Reviewer 3 Report
Much of the contents presented in the introduction section should move to a literature review section. The REVIEW work in the introduction, instead, should be focused on the MOST RELEVANT published works. The table should move to the literature review section too. Overall, the literature review section supports the research GAP in a more comprehensive and detailed way.
The hypotheses should be presented in the introduction section; a proper connection between the hypotheses and research gap/contribution(s) should be established.
The regression analysis and discussions should be restructured to address the hypotheses one by one and clearly.
Limitations of the study and the suggestions for future research should be presented in the conclusions section; different sections are unnecessary.
Overall, based on what you presented and the few VERY RELEVANT papers that you did not present, I do not see any novelties in your contribution. Your hypotheses are basically paraphrased versions of the existing works. Therefore, there are NO significant differences between your study and the existing ones.
Author Response
Reviewer #3
|
Comment |
Correction |
|
1. Much of the contents presented in the introduction section should move to a literature review section. 2. The REVIEW work in the introduction, instead, should be focused on the MOST RELEVANT published works. 3. The table should move to the literature review section too. 4. Overall, the literature review section supports the research GAP in a more comprehensive and detailed way. |
1. Heading of the literature review is removed, however, literature review starts on page 3 second paragraph. 2. Most relevant review work is added in the introduction section on page 3, highlighted in yellow colour. Also added in Table 1. 3. Moved to the literature review section. 4. Thanks for the encouraging comment. |
|
5. The hypotheses should be presented in the introduction section; a proper connection between the hypotheses and research gap/contribution(s) should be established. |
Moved in the introduction section on page 7 and highlighted in yellow colour. |
|
6. The regression analysis and discussions should be restructured to address the hypotheses one by one and clearly. |
The analysis section described the validity of hypotheses, such as Section 3.4 discusses the Hypothesis (H1), Section 3.5 addresses Hypotheses 2 and 3 (H2 and H3), Section 3.6 describe the trueness of Hypotheses 4 and 5 (H4 and H5). Section 3.7 validates hypothesis 6 (H6). However, in the discussion section, the sequence of discussion is revised according to the sequence of the hypothesis. The hypotheses are also clearly mentioned for the clear understanding of readers. These changes are highlighted in yellow colour. |
|
7. Limitations of the study and the suggestions for future research should be presented in the conclusions section; different sections are unnecessary. |
Limitations of the study and suggestions are merged as separate paragraphs in the conclusion section. The headings are removed. |
|
8. Overall, based on what you presented and the few VERY RELEVANT papers that you did not present, I do not see any novelties in your contribution. Your hypotheses are basically paraphrased versions of the existing works. Therefore, there are NO significant differences between your study and the existing ones. |
Dear Sir, we included 57 research articles relevant to the research; however, if you can specifically identify the articles, we will love to add them to the literature review. The researchers try their best to keep the work novel and are mentioned in the objectives. The perception of the manufacturing industry is majorly measured through the hypothesis. |